

# Identification of crucial genes in abdominal aortic aneurysm by WGCNA

Siliang Chen[1], Dan Yang[2], Chuxiang Lei[1], Yuan Li[1], Xiaoning Sun[1], Mengyin Chen[1], Xiao Wu[1] and Yuehong Zheng[1]

[1] Department of Vascular Surgery, Peking Union Medical College Hospital, Chinese Academy of Medical Sciences and Peking Union Medical College, Beijing, PR China
[2] Department of Computational Biology and Bioinformatics, Institute of Medicinal Plant Development, Chinese Academy of Medical Sciences and Peking Union Medical College, Beijing, PR China

Corresponding author
Yuehong Zheng,
yuehongzheng@yahoo.com

## ABSTRACT

**Background:** Abdominal aortic aneurysm (AAA) is the full thickness dilation of the abdominal aorta. However, few effective medical therapies are available. Thus, elucidating the molecular mechanism of AAA pathogenesis and exploring the potential molecular target of medical therapies for AAA is of vital importance.

**Methods:** Three expression datasets (GSE7084, GSE47472 and GSE57691) were downloaded from the Gene Expression Omnibus (GEO). These datasets were merged and then normalized using the "sva" R package. Differential expressed gene (DEG) analysis and weighted gene co-expression network analysis (WGCNA) were conducted. We compared the co-expression patterns between AAA and normal conditions, and hub genes of each functional module were identified. DEGs were mapped to co-expression network under AAA condition and a DEG co-expression network was generated. Crucial genes were identified using molecular complex detection (MCODE) (a plugin in Cytoscape).

**Results:** In our study, 6 and 10 gene modules were detected for the AAA and normal conditions, respectively, while 143 DEGs were screened. Compared to the normal condition, genes associated with immune response, inflammation and muscle contraction were clustered in three gene modules respectively under the AAA condition; the hub genes of the three modules were MAP4K1, NFIB and HPK1, respectively. A DEG co-expression network with 102 nodes and 303 edges was identified, and a hub gene cluster with 10 genes from the DEG co-expression network was detected. YIPF6, RABGAP1, ANKRD6, GPD1L, PGRMC2, HIGD1A, GMDS, MGP, SLC25A4 and FAM129A were in the cluster. The expression levels of these 10 genes showed potential diagnostic value.

**Conclusion:** Based on WGCNA, we detected 6 modules under the AAA condition and 10 modules in the normal condition. Hub genes of each module and hub gene clusters of the DEG co-expression network were identified. These genes may act as potential targets for medical therapy and diagnostic biomarkers. Further studies are needed to elucidate the detailed biological function of these genes in the pathogenesis of AAA.

## INTRODUCTION

Abdominal aortic aneurysm (AAA) is a type of true aneurysm located at the abdominal aorta, and it is defined as permanent and irreversible dilation of the abdominal aorta to 50% more than the normal aortic diameter (*Sakalihasan et al., 2018*). The prevalence of AAA in populations above 65 years old is approximately 4–8%, and it is the main cause of death in the senior population. Furthermore, the mortality of a ruptured AAA is almost 100% if left untreated (*Nordon et al., 2011*; *Sakalihasan et al., 2018*). Open surgical repair and endovascular repair remain two main methods to treat AAA. However, there are temporarily few effective medical therapies to treat AAA (*Golledge, 2019*). Therefore, elucidating the molecular mechanism of AAA pathogenesis and exploring the potential molecular targets of medical therapies for AAA is of vital importance. Although the molecular mechanism of AAA pathogenesis is not yet clear, it may involve biological processes such as immune response, chronic inflammation, oxidative stress, phenotypic switching of smooth muscle, and degradation of extracellular matrix, etc. (*Golledge et al., 2006*; *Raffort et al., 2017*; *Weintraub, 2009*). Abundant studies have been performed on the molecular mechanism of AAA pathogenesis (*Golledge, 2019*; *Raffort et al., 2017*; *Sakalihasan et al., 2018*), but weighted gene co-expression network analysis (WGCNA) has not been used to construct a gene co-expression network in AAA.

Weighted gene co-expression network analysis was developed by Horvath & Zhang in 2005 (*Langfelder & Horvath, 2008*; *Zhang & Horvath, 2005*), and an R package is available on the official website of R (https://cran.r-project.org/). WGCNA can be used to construct a weighted gene co-expression network, detect gene modules, correlate gene modules with clinical traits and identify intramodular hub genes (*Langfelder & Horvath, 2008*; *Van Dam et al., 2018*; *Zhang & Horvath, 2005*). Furthermore, WGCNA can be used not only to construct co-expression networks based on coding RNA expression data but also to construct noncoding RNA co-expression networks, such as miRNA co-expression networks and lncRNA co-expression networks (*Hu et al., 2019*; *Ma et al., 2019*).

Previous studies using microarray data of AAA have conducted differentially expressed gene (DEG) analysis and functional enrichment analysis. Gene expression profiles were compared between AAA and normal abdominal aorta, and validated genes such as CTLA4, NKTR and CD8A were identified in DEG analysis (*Biros et al., 2015*; *Lenk et al., 2007*). Nevertheless, DEGs and functional enrichment analysis cannot reveal connections and interactions among genes that are crucial in biological processes.

In our study, three Gene Expression Omnibus (GEO) mRNA microarray datasets (Table 1) were used to conduct WGCNA. Gene co-expression networks and gene modules were constructed in both AAA expression data and normal expression data. Due to the lack of clinical trait data, we used genes in all gene modules to conduct functional enrichment analysis and then mapped the DEGs to whole gene expression networks of both the AAA and normal expression data. Furthermore, we identified 10 hub genes and showed their diagnostic values by receiver operating characteristic (ROC) analysis.

**Table 1 Basic information of three datasets.**

| Year | Series | Platform | Samples (AAA:Normal) |
|------|--------|----------|---------------------|
| 2007 | GSE7084 | GPL2507 Sentrix Human-6 expression beadchip | 7:8 |
| 2013 | GSE47472 | GPL10558 Illumina HumanHT-12 V4.0 expression beadchip | 14:8 |
| 2015 | GSE57691 | GPL10558 Illumina HumanHT-12 V4.0 expression beadchip | 49:10 |

# MATERIALS AND METHODS

## Datasets

The GEO database (http://www.ncbi.nlm.nih.gov/geo/) was searched with the keywords "AAA" (Title) AND ("Series" (Entry type) AND "*Homo sapiens*" (Organism)). Finally, three datasets in the search results were included in our study: GSE7084, GSE47472 and GSE57691. Series matrix files and data tables of the microarray platform were downloaded from the GEO website.

## DEG analysis

The three series matrix files were annotated with an official gene symbol using the data table of the microarray platform, and then gene expression matrix files were obtained. The three gene expression matrix files were merged into one file, and the "sva" R package was used to conduct batch normalization of the expression data from the three different datasets. Finally, a normalized gene expression matrix file containing data from the three different datasets was obtained for DEG analysis. The "limma" R package was used to conduct DEG analysis. The threshold of DEGs was set as $|\log_2(\text{fold-change})| > 0.8$ and $p < 0.05$.

## Co-expression network construction by WGCNA

Principal component analysis (PCA) was conducted using the whole gene list and DEG list. The 2D-PCA plots showed that the variation between the AAA and normal groups was not significant (Fig. S1). Therefore, conducting WGCNA analysis using samples from AAA and normal groups separately was reasonable. Then, the "WGCNA" R package was used to construct a co-expression network for all genes in AAA and normal abdominal aorta samples. Genes with the top 25% variance were filtered by the algorithm for further analysis. Then 70 AAA samples were involved in one WGCNA analysis, while 26 normal samples were involved in another WGCNA analysis. Samples were used to calculate the Pearson's correlation matrices. Then, the weighted adjacency matrix was created with the formula $a_{mn} = |c_{mn}|^{\beta}$ (where $a_{mn}$: adjacency between gene m and gene n, $c_{mn}$: Pearson's correlation, and $\beta$: soft-power threshold). Furthermore, the weighted adjacency matrix was transformed into a topological overlap measure (TOM) matrix to estimate its connectivity property in the network. Average linkage hierarchical clustering was used to construct a clustering dendrogram of the TOM matrix. The minimal gene module size was set to 30 to obtain appropriate modules, and the threshold to merge similar modules was set to 0.25.

### Functional and pathway enrichment of gene modules

To obtain the biological functions and signaling pathways involved in each module, genes in each module were subjected to gene ontology (GO) analysis and Kyoto encyclopedia of genes genomes (KEGG) pathway analysis using The Database for Annotation, Visualization and Integrated Discovery v6.8 (https://david.ncifcrf.gov/). The threshold was set as count >2 and $p < 0.05$. To compare the two networks based on modules, the percentage of related genes in the important pathways of the modules we interested in under the AAA condition and corresponding modules under the normal condition were calculated. Chi-square tests were used to compare the percentages.

### Identification of the hub genes in functional modules and crucial gene mining

Each gene with the highest intramodular connectivity in each functional module calculated by the WGCNA algorithm was identified as a hub gene. Then, DEGs were mapped to the whole co-expression network of AAA and normal samples using Cytoscape v3.7.0, and DEG co-expression networks were obtained. The DEG co-expression network of AAA was analyzed by molecular complex detection (MCODE), a plugin in Cytoscape that clusters a given network based on topology to find densely connected regions, and the most significant cluster with 10 nodes was visualized. Then, ROC analysis was conducted with 10 genes in the cluster using SPSS 25.0.

## RESULTS

### Workflow

The workflow is shown in Fig. 1. DEG identification was conducted, and 143 DEGs were screened. WGCNA on the AAA and normal samples was performed. Co-expression networks for the AAA and normal conditions were constructed, and gene modules were also detected for further hub gene identification and comparison of expression patterns between the AAA and normal samples. Then, the DEGs were mapped to the AAA co-expression network and a DEG co-expression network was obtained. Crucial genes were identified based on this network, and ROC analysis was conducted using these crucial genes.

### Identification of DEGs

A total of 143 DEGs were identified with the threshold at $|\log_2(\text{fold-change})| > 0.8$ and $p < 0.05$ and consisted of 125 down-regulated genes and 18 up-regulated genes. The expression patterns of the DEGs are shown in Fig. 2. Down-regulated genes and up-regulated genes with a top 10-fold change ratio are shown in Table S1.

### Construction of the co-expression network of the AAA and normal conditions

The sample clustering dendrograms of the AAA and normal conditions are shown in Figs. 3A and 4A, respectively. In both conditions, all samples were included in the clusters. The soft-power threshold β was determined by the function "sft$powerEstimate"; β = 5

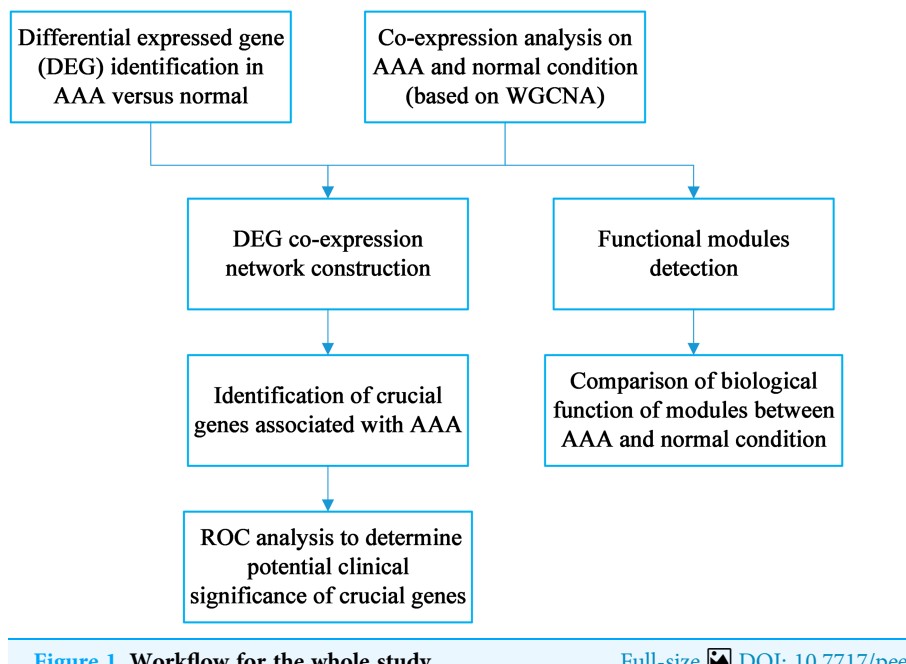

Figure 1 Workflow for the whole study.     

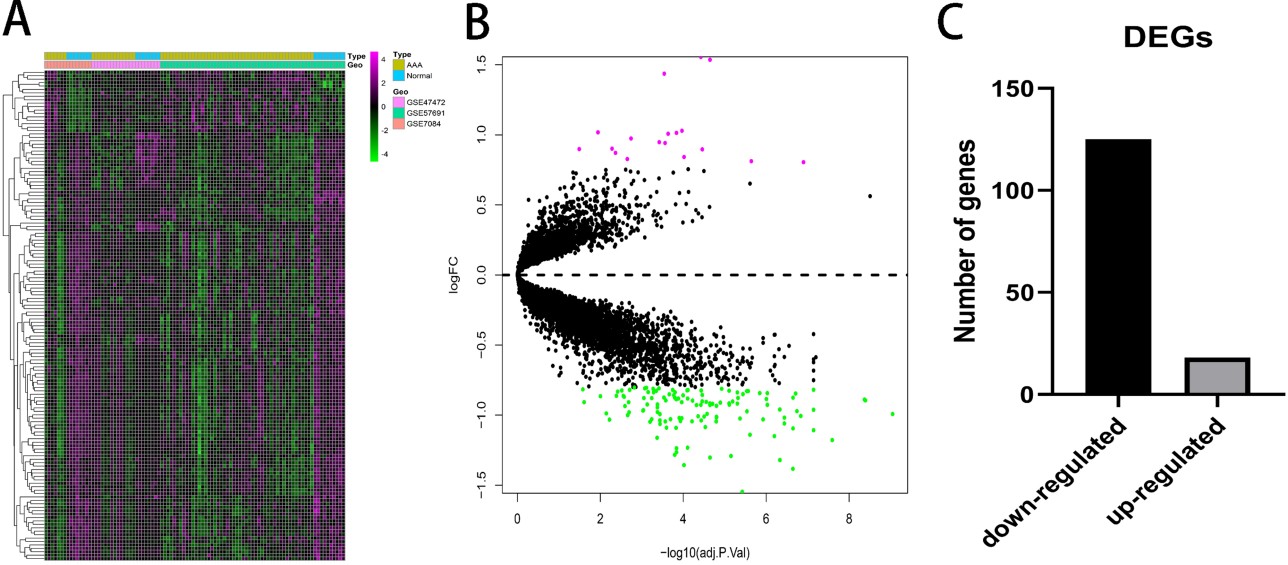

Figure 2 Differential expressed genes analysis. (A) Heatmap of 143 DEGs. The diagram presents the result of a two-way hierarchical clustering of all the DEGs and samples. (B) *X*-axis represents −log(adj.P.Val) and *Y*-axis represents logFC. Magenta dots represent genes with logFC > 0.8 and −log(adj.P.Val) < 0.05. Green dots represent genes with logFC < −0.8 and −log(adj.P.Val) < 0.05. (C) Black bar represents the number of down-regulated genes. Grey dots represents the number of up-regulated genes. 

and β = 16 were selected for further analysis of the AAA and normal conditions, respectively (Figs. 3B–3E; Figs. S2B–S2E). Then, gene modules were detected based on the TOM matrix. In the analysis, six modules were detected under the AAA condition, while 10 modules were detected under the normal condition (Figs. 3F and 3G; Figs. S2F and S2G).

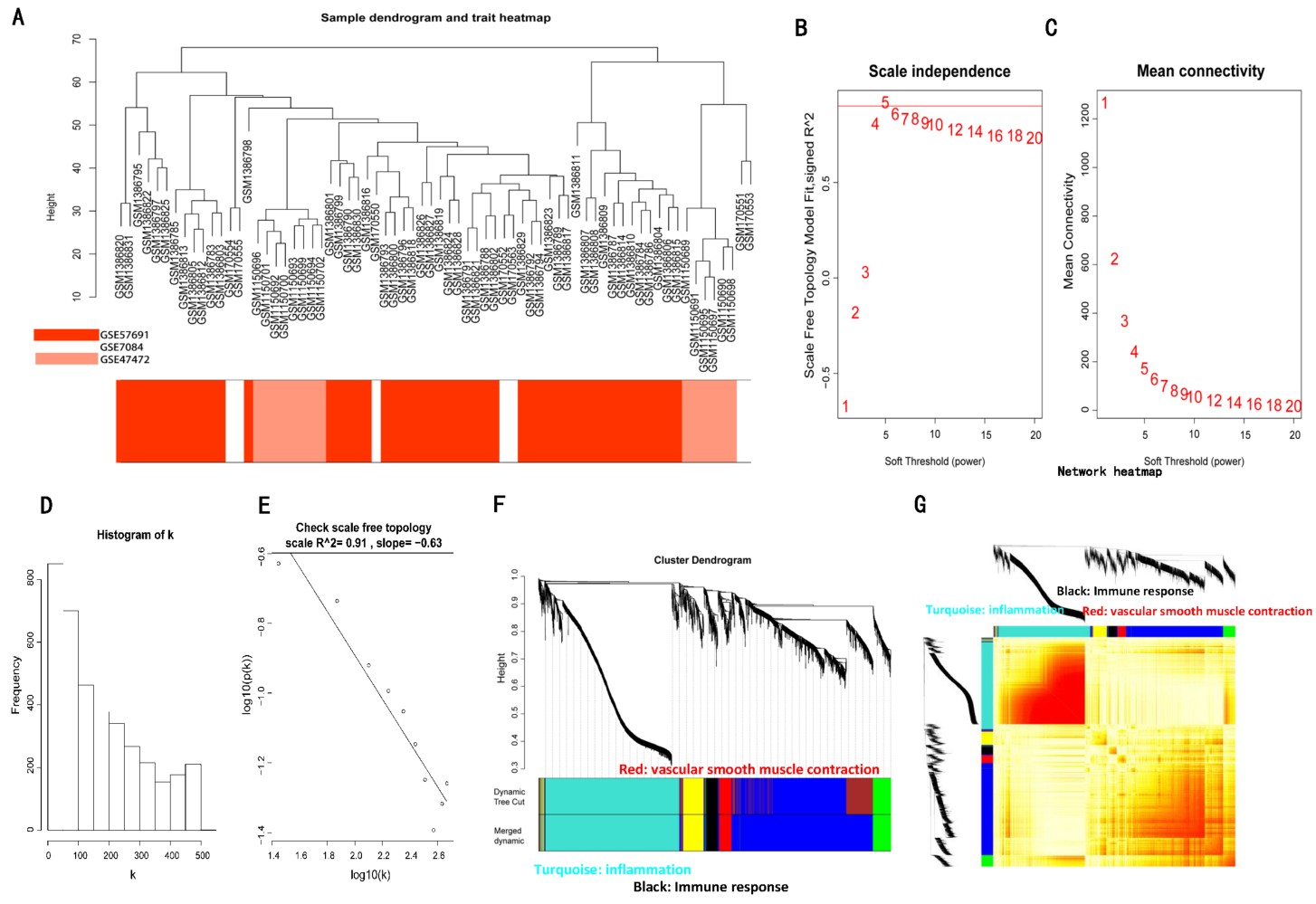

**Figure 3  WGCNA on AAA conditions.** (A) Sample clustering was conducted to detect outliers. All samples are located in the clusters and pass the cutoff thresholds. (B) and (C) Soft-thresholding power analysis was used to obtain the scale-free fit index of network topology. (D) and (E) Scale free topology when soft-thresholding power β = 5. (F) Hierarchical cluster analysis was conducted to detect co-expression clusters with corresponding color assignments. Each color represents a module in the constructed gene co-expression network by WGCNA. (G) Heatmap depicts the Topological Overlap Matrix (TOM) of genes selected for weighted co-expression network analysis. Light color represents lower overlap and red represents higher overlap.                             

## Comparison of the co-expression patterns between the AAA and normal conditions

Next, we compared the co-expression patterns between the AAA and normal conditions by examining their functional modules. We investigated the biological function of the genes in each module. GO-BP and KEGG pathway analyses were conducted. The results of the GO-BP and KEGG pathway analyses for each module are shown in Tables S2 and S3. In the network analysis, three modules in the AAA condition may be associated with the pathogenesis of AAA, and are shown in black, red and turquoise. The GO-BP terms and KEGG pathways with the top 10 counts of the three modules are shown in Fig. 4; Tables S4 and S5. The black module is mainly associated with the immune response and T/B cell activation, while the turquoise module is mainly associated with the inflammatory

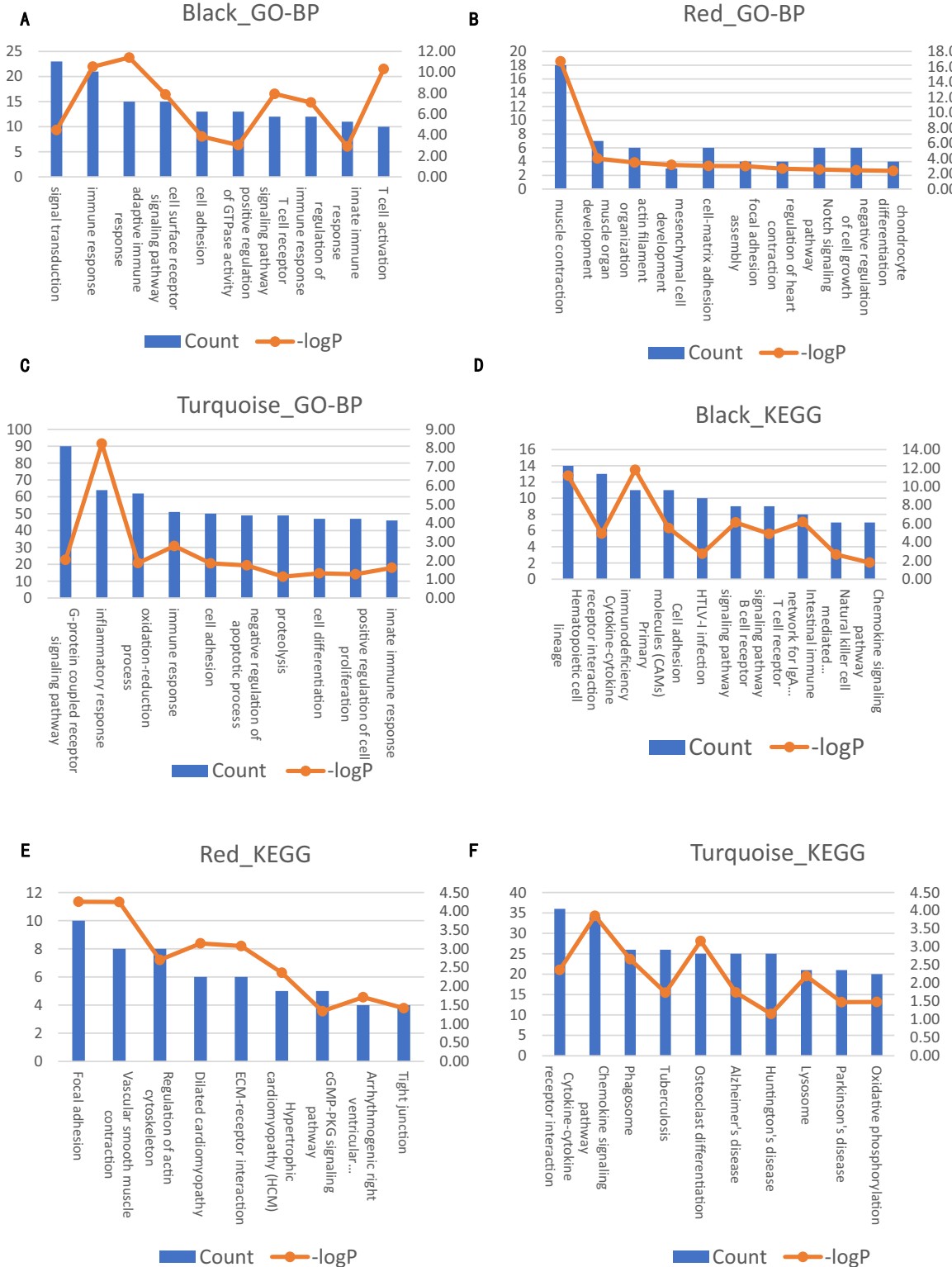

**Figure 4 Significantly enriched biological processes and KEGG pathways with top10 count number of genes in three modules on AAA condition.** The left axis represents the count number. The right axis represent the –logP. (A–C) Top10 GO-BP terms for black, red and turquoise module of AAA condition, respectively. (D–F) Top 10 KEGG terms for black, red and turquoise module of AAA condition, respectively.

**Table 2 Comparison between AAA and normal co-expression network.**

| | AAA-black $n = 50$ | Normal-darkgreen $n = 30$ | p-value | AAA-turquoise $n = 153$ | Normal-turquoise $n = 155$ | p-value | AAA-red $n = 26$ | Normal-brown $n = 49$ | p-value |
|---|---|---|---|---|---|---|---|---|---|
| Related genes % | 72.0 | 46.7 | 0.023 | 52.3 | 40.6 | 0.041 | 50.0 | 14.3 | 0.001 |
| Other genes % | 28.0 | 53.3 | | 47.7 | 59.4 | | 50.0 | 85.7 | |

response and signaling pathways in inflammation. The red module is mainly associated with muscle contraction and focal adhesion. GO-CC and GO-MF analyses were also conducted (data not shown).

In the normal condition, GO-BP terms and KEGG pathways were associated with immune response and inflammatory response in some of the modules. However, these terms and pathways were scattered in each module and are not enriched in one or two modules. Furthermore, GO-BP terms and KEGG pathways associated with muscle contraction and focal adhesion were even more scattered in each module.

Then, we compared the co-expression pattern of the two networks computationally and systematically based on the results of the KEGG pathway analysis of the gene modules. Using the black module in AAA as an example, we calculated the percentage of genes associated with immune response (hematopoietic cell lineage, primary immunodeficiency, B cell receptor signaling pathway, T cell receptor signaling pathway, intestinal immune network for IgA production, and natural killer cell mediated cytotoxicity) in all genes of the KEGG pathways with the top 10 counts (Related genes/All genes × 100%, where Related genes = the number of genes associated with targeted pathways in KEGG pathways ith the top 10 counts, and All genes = the number of all genes in KEGG pathways with the top 10 counts). We found that the pathways associated with the immune response were mainly distributed in the dark green module in the normal condition and calculated the percentage of genes associated with the immune response in all genes of the KEGG pathways with the top 10 counts (Related genes/All genes × 100%). Next, we compared the two percentages using the Chi-square test, and the percentage in the black module in the AAA condition was significantly larger than the percentage in the dark green module in the normal condition ($p < 0.05$). A similar analysis was conducted on the red and turquoise modules for the AAA condition and the corresponding modules for the normal condition. We also explained these variables in detail (Data S1). The results are shown in Table 2. These results showed that the pathways associated with inflammation, immune response and vascular smooth muscle contraction were scattered in the normal condition.

Taken together, the immune response, inflammation and vascular smooth muscle contraction play an important role in the pathogenesis of AAA.

## Identification of hub genes in the functional modules

Intramodular connectivity was calculated by the WGCNA algorithm. Genes with the highest intramodular connectivity were selected as hub genes in each module in both the AAA and normal conditions (Table S6).
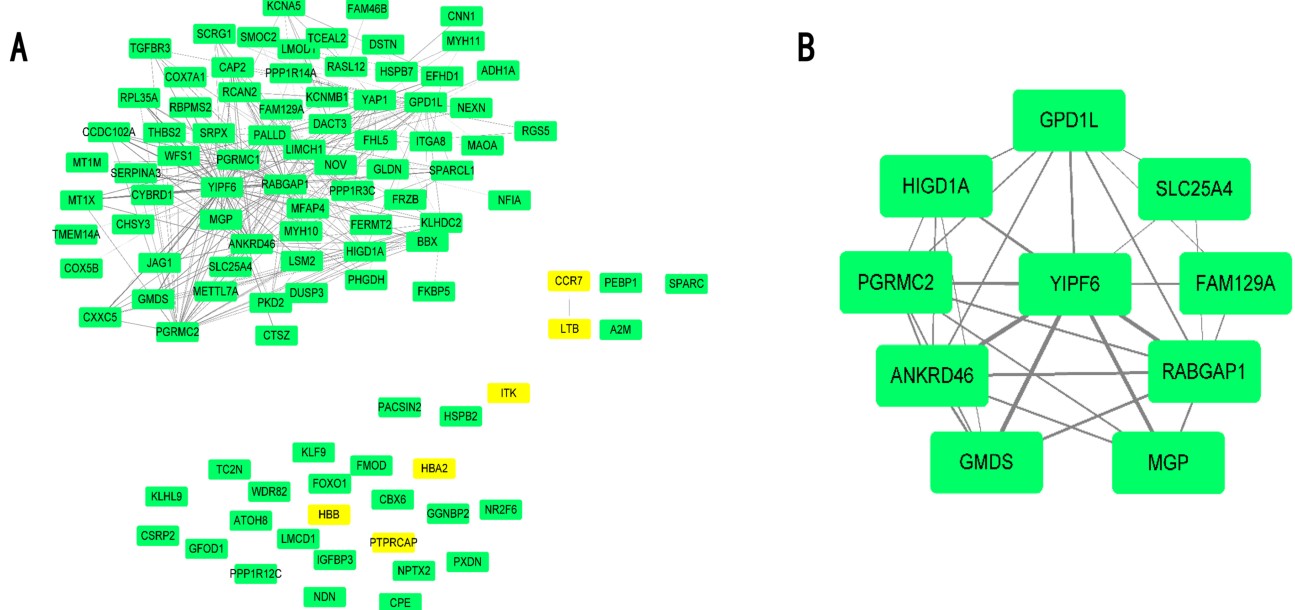

**Figure 5 DEG co-expression network and hub gene cluster.** (A) In this DEG co-expression network, Green boxes represent down-regualted genes. Light-yellow boxes represent up-regulated genes. (B) Hub gene cluster of DEG co-expression network.

The hub genes of the black, turquoise and red modules in the AAA condition were mitogen-activated protein kinase kinase kinase kinase 1 (MAP4K1), serine/threonine/ tyrosine interacting protein (STYX) and nuclear factor I B (NFIB), respectively. MAP4K1 is a regulator of JNK, immune cell adhesion and T and B cell activation (*Chuang, Wang & Tan, 2016*) and is also associated with T, B cell activation and cytokine production (*Alzabin et al., 2009*; *Shui et al., 2007*). STYX regulates apoptosis through cross-talk with FBXW7 (*Reiterer et al., 2017*). NFIB is a nuclear transcription factor, and its detailed function is still not known, but it may be associated with an immune system threshold (*Liston et al., 2012*).

## Mining crucial genes mediating AAA

Differential expressed genes were mapped to the whole co-expression network in the AAA condition using Cytoscape, and a DEG co-expression network was obtained. The threshold of weighted edges was set as 0.2 and 102 nodes, and 303 edges are in the DEG co-expression network (Fig. 5A). Then, GO-BP and KEGG analyses were conducted using genes from the DEG co-expression network, and the results are shown in Table S7. These genes were mainly associated with oxidative stress and inflammation. The most significant cluster of the DEG co-expression network detected by MCODE consisted of 10 genes and was visualized (Fig. 5B; Table 3). Yip1 domain family member 6 (YIPF6), a protein associated with clathrin-derived vesicle budding and vesicle-mediated transport, was the gene with the highest degree in the cluster.

All 10 genes were selected for ROC analysis. The area under the curve (AUC) of each gene is listed in Table 4. HIG1 hypoxia inducible domain family member 1A (HIGD1A) has the highest AUC value (AUC = 0.886, SE = 0.037, $p < 0.001$), while matrix

**Table 3 Genes in the hub gene cluster of DEG co-expression network.**

| | Gene ID | Gene symbol | Official full name |
|---|---|---|---|
| 1 | 286451 | YIPF6 | Yip1 domain family member 6 |
| 2 | 23637 | RABGAP1 | RAB GTPase activating protein 1 |
| 3 | 157567 | ANKRD46 | Ankyrin repeat domain 46 |
| 4 | 23171 | GPD1L | Glycerol-3-phosphate dehydrogenase 1 like |
| 5 | 10424 | PGRMC2 | Progesterone receptor membrane component 2 |
| 6 | 25994 | HIGD1A | HIG1 hypoxia inducible domain family member 1A |
| 7 | 2762 | GMDS | GDP-mannose 4,6-dehydratase |
| 8 | 4256 | MGP | Matrix Gla protein |
| 9 | 291 | SLC25A4 | Solute carrier family 25 member 4 |
| 10 | 116496 | FAM129A | Niban apoptosis regulator 1 |

**Table 4 The area under the curve (AUC) of 10 genes in the cluster detected by MCODE.**

| Gene symbol | AUC | $p$-value |
|---|---|---|
| YIPF6 | 0.807 | <0.001 |
| RABGAP1 | 0.797 | <0.001 |
| PGRMC2 | 0.845 | <0.001 |
| ANKRD46 | 0.832 | <0.001 |
| GPD1L | 0.831 | <0.001 |
| HIGD1A | 0.866 | <0.001 |
| GMDS | 0.845 | <0.001 |
| MGP | 0.742 | <0.001 |
| FAM129A | 0.830 | <0.001 |
| SLC25A4 | 0.811 | <0.001 |

Gla protein (MGP) has the lowest AUC value (AUC = 0.742, SE = 0.050, $p < 0.001$). The ROC curves of the genes with AUC exceeding 0.8 were plotted (Fig. 6). These genes were YIPF6, PGRMC2, ANKRD46, GPD1L, HIGD1A, GMDS, FAM129A and SLC25A4. Thus, these genes have potential diagnostic value and may become biomarkers for AAA; however, they need to be further validated in future studies.

## DISCUSSION

In our present study, 143 DEGs were screened (Fig. 2; Table S1). WGCNA was used to construct a co-expression network and detect gene modules in both AAA and normal conditions. In our analysis, six modules in the AAA condition and 10 modules in the normal condition were detected based on the co-expression network. The gene modules were subjected to GO-BP and KEGG pathway analyses, and co-expression patterns between the AAA and normal conditions were compared (Fig. 4; Tables S2 and S3). In the AAA condition, the black module was mainly associated with the immune response and T/B cell activation, while the turquoise module was mainly associated with the inflammatory response and signaling pathways in inflammation. The red module was mainly associated with muscle contraction and focal adhesion. In the normal condition,

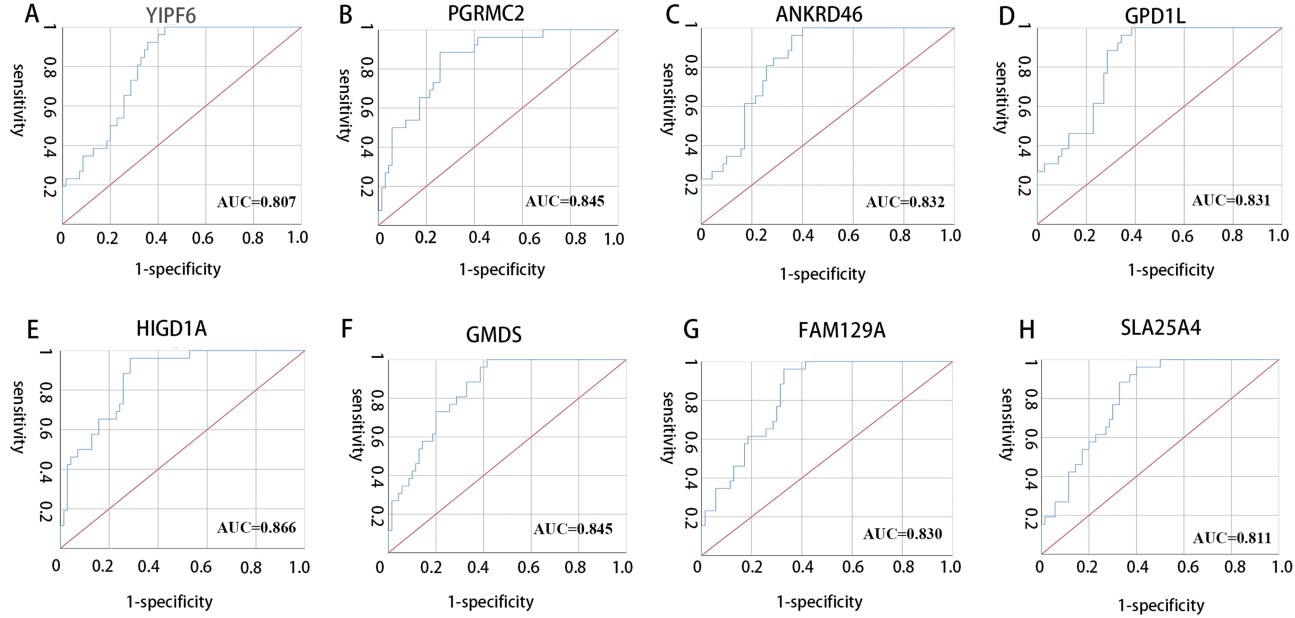

**Figure 6 ROC curves of genes with AUC exceeding 0.8.** (A–H) ROC curves for YIPF6, PGRMC2, ANKRD46, GPD1L, HIGD1A, GMDS, FAM129A and SLA25A4, respectively.

the GO-BP terms and KEGG pathways associated with immune response, inflammatory response, muscle contraction and focal adhesion were scattered in each module and were not enriched in one or two modules. These differences suggest that the immune response, inflammation and vascular smooth muscle play important roles in the pathogenesis of AAA. For each module, the genes with the highest intramodular connectivity were selected as the hub genes (Table S6). Then, DEGs were mapped to the co-expression network of AAA, and a DEG co-expression network with 102 nodes and 303 edges was obtained (Fig. 5A). The Hub gene cluster was the most significant cluster in the DEG co-expression network selected by MCODE. Ten genes were in the cluster: YIPF6, RABGAP1, ANKRD46, GPD1L, PGRMC2, HIGD1A, GMDS, MGP, FAM129A and SLC25A4 (Fig. 5B). These genes are potentially crucial genes in the pathogenesis of AAA and showed diagnostic value (Fig. 6).

A previous study using the same GEO datasets that we used screened 1,199 DEGs (*Wan et al., 2018*), which is far more than the number of DEGs we screened. However, unlike the previous study, we used $\log_2|\text{FoldChange}| > 0.8$ as another threshold, and the adjusted *p*-value was used. Furthermore, the "sva" R package was used to normalize the three datasets before DEG analysis in our study. The differences in choosing parameters and normalization may account for the different number of DEGs screened.

Previous genome-wide association studies (GWASs) reported some significant AAA-associated Single-nucleotide polymorphism (SNPs) and genes from genomic data. *Helgadottir et al. (2008)* discovered one common variant (rs10757278-G on chromosome 9p21, OR = 1.31, $p = 1.2 \times 10^{-12}$) that is associated with the risk of AAA. Other SNPs, such as rs7025486 on 9q33 in the DAB2IP gene (OR = 1.21 and $p = 4.6 \times 10^{-10}$) (*Gretarsdottir et al., 2010*), rs1466535 on 12q13.3 in the LRP1 gene (OR = 1.15 and

$p = 4.52 \times 10^{-10}$) (*Bown et al., 2011*), rs599839 on 1p13.3 in the SORT1 gene (OR = 0.81 and $p = 7.2 \times 10^{-14}$) (*Jones et al., 2013*) and rs6511720 in LDLR (OR = 0.76, $p = 2.08 \times 10^{-10}$) (*Bradley et al., 2013*), are significantly associated with AAA. *Jones et al. (2017)* conducted a meta-analysis of GWAS for AAA and identified 1q32.3 (SMYD2), 13q12.11 (LINC00540), 20q13.12 (near PCIF1/MMP9/ZNF335), and 21q22.2 (ERG) as four new loci associated with AAA and found that these four loci may be specifically associated with AAA compared with other cardiovascular diseases. We screened DEGs based on transcriptomic data, and there was some overlap between previous GWASs and DEGs we screened. MMP9 is in the DEG list with a fold-change of 2.03, which is in accord with the study by *Jones et al. (2017)* which found that one SNP near PCIF1/MMP9/ZNF335 was significantly associated with AAA. Furthermore, one SNP in LRP1 was associated with AAA, and *Boucher et al. (2007)* found that the LRP1 gene plays a protective role in vascular disease via TGF-β signaling (*Bown et al., 2011*). Thus, the TGFBR3 gene with a fold-change of −1.01 in our DEG list may be associated with LRP1 gene function and pathogenesis of AAA, although the detailed mechanism has not been clearly elucidated. Furthermore, we obtained genes closest to SNPs in those literature, respectively (i.e., GWAS genes) (Table S8) using dbSNP (https://www.ncbi.nlm.nih.gov/snp/) and found that there were two GWAS genes (GSN and MMP9) overlapped with DEGs. Then we compared the GWAS genes with genes enriched in GO-BP and KEGG pathway terms of DEGs in the co-expression network (Table S7) and there were also no overlap. Therefore, we inferred that GWAS genes were not "close" to DEGs in the co-expression network in the functional level. We also evaluated the association between SNPs and DEGs using eQTL calculator in GTEx website (https://gtexportal.org/home/testyourown) (*Brown et al., 2017*). Although some normalized effect size between some SNPs and genes were significant (Table S9), the biological significance is not clear and further validation is needed. Some researchers combined genomic data and patient history of AAA. Li et al. integrated genomic data of AAA based on whole genome sequencing and electronic health record data using a machine learning approach. Their study provided a paradigm for both understanding the pathogenesis of complex diseases such as AAA and managing the personal health of patients (*Li et al., 2018*). *Duren et al. (2017)* developed a novel method (paired expression chromatin accessibility) to construct enhancer gene network and enable us to view how the regulatory elements works. Both of these studies provided future directions for studying the pathogenesis of AAA.

The comparison of gene modules based on GO-BP and KEGG between the AAA and normal conditions indicates that these two conditions have different co-expression patterns. Using RNA microarray, *Spin et al. (2011)* showed that in AAA mice, immune genes were widely upregulated. Furthermore, evidence for the presence of innate and adaptive immune systems in the pathogenesis of AAA has been provided by many studies (*Coscas et al., 2018*; *Lareyre et al., 2017*; *Senemaud et al., 2017*; *Trachet et al., 2016*). Our finding that the immune response was clustered in the black module is consistent with the studies mentioned above. Inflammation also plays an important role in the pathogenesis of AAA. In addition, *Spin et al. (2011)* found that inflammatory genes are also up-regulated in AAA mice. Cells related to inflammation such as B cells, T cells, natural

killer cells and macrophages and their products have been found in AAA tissue (*Biros et al., 2014; Furusho et al., 2018; Meng et al., 2016; Tsuruda et al., 2008; Zhang et al., 2011*). Studies have also shown that regulatory T cells can inhibit AAA development in human and animal models such as mice by inhibiting the inflammatory response in AAA (*Ait-Oufella et al., 2013; Yodoi et al., 2015; Zhou et al., 2015*), which further supports the inflammation-mediated theory of AAA. Our findings on the turquoise module agree with the results reported by these studies. Our study has also shown that vascular smooth muscle contraction is associated with AAA pathogenesis. Unlike the immune response and inflammation, little is known about the relationship between vascular smooth muscle contraction and the pathogenesis of AAA. However, *Chew, Conte & Khalil (2004)* and *Chew, Orshal & Khalil (2003, 2004)* found that elastase and matrix metalloproteinase (MMP, mainly MMP2 and MMP9) both promote early dilation of AAA by inhibiting vascular smooth muscle contraction through the mechanism of $Ca^{2+}$ entry inhibition of vascular smooth muscle. Based on the GO-BP and KEGG pathway analyses, we found that the genes in the DEG co-expression network were mainly associated with oxidative stress and inflammation. *Emeto et al. (2016)* summarized that inflammation, oxidative stress and reactive oxygen species play an important role in the pathogenesis of AAA and believed that oxidative stress induces inflammation and that tissue injury is caused in the pathogenesis of AAA.

The hub genes of the black, turquoise and red modules in the AAA condition were MAP4K1, STYX and NFIB, respectively. MAP4K1, also named hematopoietic progenitor kinase 1, is expressed in T cells, B cells, and macrophages, as well as hematopoietic progenitor cells. It serves as a regulator of JNK and T and B cell signaling. It can also regulate immune cell adhesion (*Chuang, Wang & Tan, 2016*) and is associated with T and B cell activation and cytokine production (*Alzabin et al., 2009; Shui et al., 2007*), which is consistent with our findings in the functional enrichment analysis of the black module in the AAA condition. STYX is a pseudophosphatase with no actual catalytic activity. It serves as an anchor for MAPKs and plays an important role in their function (*Reiterer, Pawlowski & Farhan, 2017*). In addition, STYX regulates apoptosis in breast cancer cells and HeLa cells through cross-talk with FBXW7 (*Reiterer et al., 2017*). Therefore, STYX may also play a role in regulating vascular smooth muscle apoptosis in the pathogenesis of AAA. NFIB is a nuclear transcription factor; however, its detailed functions are still not known (*Ono & Okada, 2018*). It can be downregulated by microRNA-29, which is an immune system threshold. In conclusion, these three hub genes from the three key WGCNA gene modules may be of vital importance in the pathogenesis of AAA.

Yip1 domain family member 6 was the gene with the highest degree in the hub gene cluster generated from the DEG co-expression network. YIPF6 is a member of the Yip1 family of proteins and is located in the Golgi apparatus. It is believed to participate in vesicular transport (*Kranjc et al., 2017*). *Brandl et al. (2012)* discovered that one type of null mutation of YIPF6 induces spontaneous intestinal inflammation in mice, but the mechanism seems to be related to granule secretion by Paneth and goblet cells. Whether YIPF6 is associated with chronic inflammation in AAA is unknown, but it may play a role in the balance between proinflammatory and anti-inflammatory cytokine production.

However, the ROC curve showed that YIPF6 as well as other genes in this cluster has potential diagnostic value.

In this study, we used WGCNA to construct a co-expression network, detect gene modules and identify hub genes in AAA for the first time. In addition, we merged and normalized three GEO datasets instead of using only one to conduct our analysis. However, due to the lack of clinical trait data in the three GEO datasets, we did not correlate gene modules with clinical traits when conducting the WGCNA.

Here, our study revealed that the co-expression patterns between the AAA and normal conditions are different. These differences indicate that immune response, inflammation and vascular smooth muscle contraction may play important roles in the pathogenesis of AAA. Then, we identified hub genes of gene modules in both the AAA and normal conditions and hub gene clusters of the DEG co-expression network. These crucial genes may have crucial biological functions in the pathogenesis of AAA.

## CONCLUSIONS

Based on WGCNA, we detected six modules in the AAA condition and 10 modules in the normal condition. Hub genes of each module and hub gene clusters of the DEG co-expression network were identified. These genes may act as potential targets for medical therapy and diagnostic biomarkers. Further studies are needed to elucidate the detailed biological function of these genes in the pathogenesis of AAA.

### Funding

This work was supported by the Natural Science Foundation of China (81770481 and 51890894), the Natural Science Foundation of Beijing (7172171) and the CAMS Innovation Fund for Medical Sciences (CIFMS, 2017-I2M-1-008). The funders had no role in study design, data collection and analysis, decision to publish, or preparation of the manuscript.

### Grant Disclosures

The following grant information was disclosed by the authors:
Natural Science Foundation of China: 81770481 and 51890894.
Natural Science Foundation of Beijing: 7172171.
CAMS Innovation Fund for Medical Sciences (CIFMS, 2017-I2M-1-008).

### Competing Interests

The authors declare that they have no competing interests.

### Author Contributions

- Siliang Chen conceived and designed the experiments, performed the experiments, analyzed the data, contributed reagents/materials/analysis tools, prepared figures and/or tables, authored or reviewed drafts of the paper, approved the final draft.

- Dan Yang conceived and designed the experiments, performed the experiments, analyzed the data, authored or reviewed drafts of the paper, approved the final draft.
- Chuxiang Lei analyzed the data, authored or reviewed drafts of the paper, approved the final draft.
- Yuan Li analyzed the data, prepared figures and/or tables, approved the final draft.
- Xiaoning Sun analyzed the data, authored or reviewed drafts of the paper, approved the final draft.
- Mengyin Chen analyzed the data, prepared figures and/or tables, approved the final draft.
- Xiao Wu analyzed the data, prepared figures and/or tables, approved the final draft.
- Yuehong Zheng conceived and designed the experiments, authored or reviewed drafts of the paper, approved the final draft.

## Data Availability

Data is available at NCBI GEO: GSE7084, GSE47472 and GSE57691.

## Supplemental Information

Supplemental information for this article can be found online at http://dx.doi.org/10.7717/peerj.7873#supplemental-information.

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
