# Peer review of "Identification of crucial genes in abdominal aortic aneurysm by WGCNA"

_PeerJ, doi:10.7717/peerj.7873_

## Round 0.1 · original submission · Major Revisions

The reviewers are concerned the connection with current GWAS study for AAA. Also the presentation needs to be improved.

[]

Reviewer 1 ·

Basic reporting

In this paper, the authors first detected differentially expressed genes between AAA samples and controls. Then they used WGCNA to construct co-expression networks to identify key modules and hub genes. The following lists my major concerns.

Experimental design

no comment

Validity of the findings

no comment

Additional comments

1. The figures in this manuscript are not clear, especially for Figures 3, 4, 5, and 6. The characters are not clear to read. Can you draw them with higher quality?
2. In line 119, the authors used 0.8 as fold change threshold. Why did you use 0.8 (2^0.8~=1.7)? Does this cutoff influence your analysis?
3. In figure 3D and 3D, you used WGCNA to identify modules. Can you annotate the clusters with functional terms? Your figures could not deliver enough information.
4. Similar to above question, in figure 6, you only show the down- and up-regulated genes. What are related functions of these genes?
5. From line 176, the authors introduce the method to compare the AAA and control co-expression networks. Basically, the two networks are different in nodes and edges. Do you have computational method systematically compare them? In your result, the pathways like inflammation is appeared in both networks, although, it is scattered in one module. How can you tell they are different on network level?
6. There are some GWAS studies reported some significantly AAA associated SNPs and genes from genomic data. Can you have a comparison between your differentially expressed gene list with them? Can your co-expression network explained the disease mechanism from genetic variations?

Reviewer 2 ·

Basic reporting

'no comment'

Experimental design

'no comment'

Validity of the findings

'no comment'

Additional comments

The authors analyze three gene expression datasets on both normal and AAA conditions. Network analysis is the best way to identify crucial genes in disease. Overall, the workflow for the whole study is reasonable, with some important caveats outlined below. As a whole, the manuscript would be more compelling if it combine the network analysis with GWAS study.

1, There are several GWAS studies on AAA. Following papers found some risk genes for this disease. I think it will be great to combine the current network analysis with GWAS study.
Jones, Gregory T., et al. "Meta-analysis of genome-wide association studies for abdominal aortic aneurysm identifies four new disease-specific risk loci." Circulation research 120.2 (2017): 341-353.
Li, Jingjing, et al. "Decoding the genomics of abdominal aortic aneurysm." Cell 174.6 (2018): 1361-1372.

2, I guess the variation between AAA and normal groups are big compared to the within group variation. Authors should show the variation of samples by Principal component analysis (PCA) based on 1) whole gene list and 2) differential gene list. If the variation between two groups are big, it is good to merge all the samples together and do network analysis. Only using the samples from normal condition and constructing network would loss too much information. Modules on this network (from all samples) have different expression pattern, some of the modules would have high expression on AAA condition while some have high expression on normal conditions.

3, Please improve the quality of figures and enlarge the font size. It is hard to see the text on figures after zoom in 500%. Fig.2A, Fig.3A, Fig. 4A, Fig. 6A. In Fig.6, don’t use the color of red.

4, In the section “Mining crucial genes mediated in AAA”, please give detail of the ROC analysis. What is the features, what is the ground truth,…

5, please improve the language and writing.
i.e. “One hundred and two nodes and three hundred and three edges” > “102 nodes and 303 egdes”

---

## Round 0.2 · Minor Revisions

The reviewers request further discussions to improve the manuscript. I suggest minor revision before acceptance.

Reviewer 1 ·

Basic reporting

Null

Experimental design

Null

Validity of the findings

Null

Additional comments

Thanks for the authors' response. I have the following concerns.

1. About the quality of figures, I checked the original PNG files. Although the resolution is enough for clear visualization after zooming out, the character size is still not large enough when printing these figures on an A4 size paper. So I strongly suggest to improve the visualizing effect of all figures when reaching to the final publishing round.

2. Figures 3F,G and 4F,G show the clustering modules by WGCNA, but the figures are not informative. I think labeling the exact functional terms you listed in Table S2-3 aside the modules will directly convey the information and improve the figures. Figure 3 and 4 are similar and redundant, it’s not clear enough for readers to tell the similarity and difference between normal and AAA. What do you want to deliver? Do you have better way to directly show? Otherwise you can put some of the figure to supplementary files.

3. For last question 5, i.e. “Do you have computational method systematically compare them? In your result, the pathways like inflammation is appeared in both networks, although, it is scattered in one module. How can you tell they are different on network level? “. I could not understand your computation method. You mentioned “Related genes=Number of genes associated with targeted pathways in KEEG pathways with top 10 count number, All genes=Number of all genes in KEGG pathways with top 10 count number) ”. What does this mean? Can you explain it clearly and detially?

4. About the GWAS study, you mentioned only two genes have some consistent evidence. What about the others? Does this mean there is still a big gap between the genetic variations and co-expression network? Do you have any comment on it?

Reviewer 2 ·

Basic reporting

none

Experimental design

none

Validity of the findings

none

Additional comments

The authors’ response solved most of my concerns. But I still have some suggestions:
1, About the combination with GWAS study, I think the authors should go deeper. Those GWAS study provide some significant SNPs, which can be assigned to the genes by distance, enhancer-gene network (by PECA, http://web.stanford.edu/~zduren/PECA/), or eQTL (by GTEx, https://gtexportal.org/home/). Some interesting questions are:
1) Are those GWAS genes are significantly overlapped with the DEGs?
1) Are those GWAS gene close to the DEGs in co-expression network?
PECA paper: Duren, Zhana, et al. "Modeling gene regulation from paired expression and chromatin accessibility data." Proceedings of the National Academy of Sciences 114.25 (2017): E4914-E4923.
eQTL paper: Brown, Andrew Anand, et al. "Predicting causal variants affecting expression by using whole-genome sequencing and RNA-seq from multiple human tissues." Nature genetics 49.12 (2017): 1747.

2, About the variation between AAA and normal, I think the PCA plots in response letter are informative. Because the variation between AAA and normal is small, so you need to 1) collect multiple gene expression data to differential analysis, 2) use network analysis to de-noise. I suggest authors put those figures to the article and show the variation first. It is a good motivation for collecting multiple datasets and network analysis.

3, I download the raw figures and find the quality is good. But I think authors need to enlarge the font size to make those easier to read. For example, the text in the Figure3 A are too small to read (please print it on an A4 size paper and try to read it). In Figure 3A, it is good to show the samples’ conditions (AAA or normal) and source study (GSE7084, GSE47472 and GSE57691) by different colors or markers.

---

## Round 0.3 · accepted · Accept

Both two reviewers are satisfied with the revision and agree it improves the previous version. I recommend its acceptance.

Reviewer 1 ·

Basic reporting

Null

Experimental design

Null

Validity of the findings

Null

Additional comments

Thanks for the authors' detailed response. I have no further question or comment.

Reviewer 2 ·

Basic reporting

None

Experimental design

None

Validity of the findings

None

Additional comments

The authors have addressed my comments.